# Dairy Farmers’ Perceptions of and Actions in Relation to Lameness Management

**DOI:** 10.3390/ani9050270

**Published:** 2019-05-23

**Authors:** Mohammed Babatunde Sadiq, Siti Zubaidah Ramanoon, Wan Mastura Shaik Mossadeq, Rozaihan Mansor, Sharifah Salmah Syed Hussain

**Affiliations:** 1Department of Farm and Exotic Animal Medicine and Surgery, Faculty of Veterinary Medicine, Universiti Putra Malaysia, Serdang 43400 UPM, Selangor, Malaysia; sadiquemohammed99@yahoo.com (M.B.S.); rozaihan@upm.edu.my (R.M.); 2Research Centre for Ruminant, Faculty of Veterinary Medicine, Universiti Putra Malaysia, Serdang 43400 UPM, Selangor, Malaysia; 3Department of Veterinary Preclinical Sciences, Faculty of Veterinary Medicine, Universiti Putra Malaysia, Serdang 43400 UPM, Selangor, Malaysia; wmastura@upm.edu.my; 4Department of Veterinary Clinical Studies, Faculty of Veterinary Medicine, Universiti Putra Malaysia, Serdang 43400 UPM, Selangor, Malaysia; ssalmah@upm.edu.my

**Keywords:** animal welfare, lameness, farmer, dairy cows, perception

## Abstract

**Simple Summary:**

Lameness is a pressing issue in dairy production. Dairy farmers are primarily responsible for the welfare of their cows and decision-making regarding lameness management. However, there are concerns regarding the communication of the importance of lameness to farmers and their motivation towards proper management. A review of the literature indicates that various factors influence farmers’ perceptions about lameness, their adoption of recommended measures and whether or not they end up treating a lame cow. This review summarizes these related issues in consideration of the welfare and economic implications of farmers’ decisions. The information herein is vital to the identification of measures on how to motivate dairy farmers towards appropriate lameness management.

**Abstract:**

Lameness continues to be a welfare and economic issue for dairy cows. However, the consequences of lameness seem to be better understood by veterinarians and related personnel in comparison to dairy farmers. Prompt detection and treatment of lame cows is essential in reducing its negative impact on milk processing systems. To that end, understanding farmers’ perceptions regarding the significance of lameness to dairy cows is vital. One fundamental aspect is the underestimation of lameness prevalence by dairy farmers, which is as a result of different understanding of the problem. The same applies to their decision to treat lame cows and to adopt various detection and management practices. All of these shortcomings contribute to poor cattle welfare and economic losses in dairy production. This review summarizes the results of studies that have investigated dairy farmers’ perceptions of lameness and the associated implications on the wellbeing and productivity of dairy cows. Factors associated with farmers’ attitudes toward claw health and lameness management are also presented. Additionally, economic observations relating to lameness prevention, treatment and the adoption of lameness detection systems are also highlighted. To strengthen these points, interventional programmes requiring farmers’ participation are discussed as a promising approach in answering some of these challenges. A review of the literature indicates both the opportunities and barriers inherent in the tackling the lameness issue from the farmers’ perspectives. Such knowledge is crucial in identifying measures on how to motivate dairy farmers towards proper lameness management.

## 1. Introduction

Lameness causes pain and behavioral changes in dairy cattle. It also affects milk production due to the stress caused to the animals [1]. Therefore, lameness remains an economic problem for milk production systems [2]. Several papers have reported the occurrence of dairy cow lameness in various parts of the globe and its multifactorial etiology [1,3]. Likewise, recent studies have highlighted the importance of various lameness detection systems and treatment methods for specific claw lesions [2,3]. However, despite the substantial work in lameness related research, there are still concerns regarding the increasing occurrence of lameness in dairy herds.

Farmers, veterinarians and dairy consultants are primarily responsible for the welfare of dairy animals and decision-making concerning claw health. Thus, effective communication between them is necessary for improved claw health. Farmers’ perceptions regarding lameness, their understandings of what constitutes a lame cow and knowledge regarding the associated factors or practices related to claw health have been investigated in several studies [4,5]. While some farmers viewed lameness as an important health problem [4], others held a contrary opinion [6,7]. Such differing perceptions are influenced by several factors [4,6,7]. For instance, the farmers interviewed by Olmos et al. [6] were of the opinion that pasture-based farming provides an optimal condition for claw health, thus they were not concerned about lameness as a problem for their herds. Likewise, farmers tended to associate pasture access with good animal welfare [8]. The importance attached to lameness by farmers could either be based on their general perceptions of the problem or related to the perceived gravity of the problem for their herds [4]. To that effect, the underestimation of lameness prevalence by farmers, in comparison to that of trained personnel, remains significant [5]. These inaccurate perceptions result in late lameness detection, increased incidence of chronically lame cows and increased economic loss [2].

Recent studies have also advised that dairy farmers need to be educated on the economic implications of lameness, so as to encourage them to adopt recommended measures [2,7]. This includes an assessment of the cost of clinical lameness or specific claw lesions on productivity [9,10], as well as the treatment and preventive costs related to lameness detection systems [2,11]. Hence, information on the welfare and economic implications of lameness could be synergized accordingly. Another area of interest is farmers’ participation in both the assessment and implementation of strategies to improve claw health. Such evidence-based approaches depict the role of farmers in tackling the lameness problem [5].

Unfortunately, there is no current data that summarizes the findings and extent of the research that has been conducted regarding farmers’ perceptions of lameness and factors influencing their lameness management practices. This issue is significant to veterinarians and related health staff to understand the barriers to effective lameness control, as well as to motivate farmers to implement recommended strategies. A systematic review of the literature is a well-known means of identifying knowledge gaps and providing direction for future research [12]. The objective of this review is to evaluate the perceptions of dairy farmers towards lameness, identify the barriers to its effective management and suggest ways of motivating dairy farmers to take action.

## 2. Methods

The approach used in this review consists of a narrative integrative style and basic aspects of a systematic review in order to minimize bias, as described by Sargeant and O’Connor [12]. The papers included in this review fulfilled the following criteria: focused on lameness-related research on adult dairy cows, focused on the involvement of farmers or veterinarians in the data collection, highlighted either welfare or economic problems in lameness management, was written in English and was published in a peer-reviewed journal. The literature search was conducted from March to October 2018 and the databases used included PubMed, Google scholar and Web of Science. Papers that were cited in the works that were obtained from the databases were also assessed for inclusion. The search terms used included ‘dairy,’ ‘farmer,’ ‘claw lesions,’ ‘lameness’ and ‘welfare.’ A total of 793 records were retrieved from the 3 aforementioned databases, while 55 cited papers were assessed from the original papers. The literature selection process involved the use of the Preferred Reporting Items for Systematic Reviews and Meta-analyses (PRISMA) checklist [13].

Of the 793 records, 260 duplicates were removed and the remaining 533 were further reviewed.

Upon re-evaluation, 40 papers were within the inclusion criteria and the main objectives and findings were used to categorize the results under six main topics: the importance of lameness to the dairy farmer (number of papers = 10), detection of lame cows (*n* = 9), definition of lameness and knowledge factors influencing farmers’ decisions on the management of lame cows (*n* = 13) and economic related aspects (*n* = 9).

## 3. Importance of Lameness to the Dairy Farmer

Several studies have investigated how dairy farmers view the problem of lameness. For instance, with regard to animal-based measures, the four main health problems mentioned by Italian farmers included lameness and hoof lesions, records of mortality among adult cows and calves, observation of integument alterations and low body condition score (BCS) [14]. A study by Martin-Collado et al. [15] assessed preferences among farmers for cow trait improvements. Lameness was the main health problem that required urgent attention in the participants’ farms, followed by mastitis and calving difficulty [15]. Similarly, farmers in Austria mentioned lameness and other reproductive disorders as the most crucial challenges in their herds [16]. The findings from other surveys ranked lameness second in importance as a major health problem in dairy cows, after mastitis [4,17]. However, farmers in a large Canadian study ranked lameness ahead of mastitis [18]. These studies exemplify the significance of the condition in various parts of the globe.

The factors influencing farmers’ perceptions of health issues include their definition of the health problem, their educational status and their farming experience [19]. Farmers might perceive lameness as a problem despite its not being a current health issue in their herds. Such a perception often emanates from formal education, training and past experience of lameness cases on their farms [4,20]. In some cases, current on-farm prevalence may convince less well-informed farmers of the gravity of the problem [21]. These events have been shown to shift farmers’ attitudes towards adopting lameness prevention and management strategies [4,7].

The prevalence of other diseases may also affect farmers’ perceptions of lameness and claw health. In conditions where the farmer has observed other health issues to be more prevalent, lameness is perceived as a more minor problem [4,21]. Again, most farmers in the study of Bruijnis et al. [7] were satisfied with the claw health of their cows and were unlikely to take any related action, as they would with other health issues. However, farmers’ inability to detect a lame cow and underestimation of the prevalence of lameness may cloud their perception of the real on-farm situation [6,21].

It has been suggested that the welfare of lame cows greatly influences farmers’ perception of the problem [4]. Quite a number of authors recommend that educating farmers about the welfare implications in lame cows is more likely to encourage farmers to take action [4,17,22]. Results from two studies showed that farmers perceived that lameness causes pain [4,17] and Leach et al. [4] reported that farmers felt sorry for the affected cows. Also, using a pain scale, majority of the farmers who selected higher scores on the scale advised for pain management when treating lame cows [22]. Nevertheless, 25% of the farmers surveyed by Bruijnis et al. [7] thought that lame cows do not suffer any pain and that the measures taken to improve animal welfare were unrelated to claw health.

## 4. Detection of Lame Cows

The ability of a farmer to detect lame cows is vital in understanding the severity of the problem [5]. Also, early treatment of clinical cases is fundamental in reducing lameness prevalence in dairy herds [23,24]. However, studies have shown that farmers often underestimate lameness prevalence in their herds compared to trained personnel [25,26]. Dahl-Pedersen et al. [27] in a recent study enrolled farmers, veterinarians and livestock drivers to score lameness using video recordings. The farmers’ group had the lowest level of intra-rater agreement compared to the other participating groups [27].

Various studies regarding farmers’ detection and estimation of lameness prevalence are summarized in Table 1. The prevalence of lameness as reported by farmers ranged from 0 to 20%, whereas that of trained personnel ranged from 1.2 to 64%. Thus, the majority of farmers were about three times less likely than skilled personnel to detect the lame cows. Two studies reported positive correlations between farmers’ and researchers’ prevalence estimates [20,21]; nevertheless, in the former, the studied herds were managed in different settings. In a study by Bran et al. [21], similar prevalence estimates between farmers and the researcher was recorded for only severely lame cows. In the same study, the veterinarian’s estimate of lameness prevalence was 10% higher in farms where farmers identified lameness as a primary health problem (40%), compared to farms where farmers did not recognize lameness as a problem [21]. However, caution is needed when comparing the findings from the various studies, owing to differences between them in the locomotion scoring scales (LSSs) employed for lameness detection and management systems.

The treatment of lame cows also demands proper understanding of the specific causes. An important finding following a survey of farmers on the on-farm treatment of claw horn lesions indicated that they were unable to differentiate between mild sole lesions, sole ulcers (SU) and white line disease (WLD) [31]. Bran et al. [21] reported that smallholder farmers in southern Brazil admitted the frequent use of topical hoof products and antibiotics in the treatment of lame cows, with limited application of claw trimming (CT) and analgesics/anti-inflammatories. Recently, about 58% and 42% of the farmers requested to detect cows with digital dermatitis (DD) in their herd either underestimated or overestimated the prevalence, respectively, compared with that of the researcher [26].

## 5. Definition of Lameness, Related Terms and Knowledge About the Risk Factors

One of the key issues relating to the perception of farmers about lameness is vague and poor definitions of the concept. This problem was reported to influence both the detection of lame cows and decision-making on when and whether a cow should be treated [5,6]. In this regard, the language used in describing lameness to the farmer is vital. According to Horseman et al. [4], the underestimation of lameness by farmers may be the result of their interpretation of what they see as lameness. Trained personnel can detect mild to moderate lameness, which are the cows classified as having a locomotion score (LS) of 2 (based on the 5-point LS developed by Sprecher). However, farmers may not perceive such an animal as lame, since they often reserve the term ‘lame’ for severely affected cows (LS 4 and 5) [4]. The fact that the LS classifies animals into various degrees of lameness remains difficult for the farmer to comprehend. Nevertheless, the prompt detection and treatment of mildly lame cows is critical in preventing complications and promoting recovery [10,32].

Results from a recent survey by Olmos et al. [6] showed that the participants (farmers and dairy consultants), in describing gait alteration, preferred to use terms related to hoof problems or laminitis and they were of the opinion that such disorders were normal and required no intervention. However, laminitis was frequently perceived as closely related to lameness [6]. The fact that the respondents had no idea of the connection between lameness and hoof lesions is evidence of their poor understanding of ‘lameness.’ Additionally, one consequence of such uncertainty is the variation in management plans drawn up to address the situation. Depending on the perceived risk of lameness as caused by laminitis or hoof disorders, farmers select different preventive measures [6]. Therefore, there is a disconnection between the definition of lameness and identification of a lame cow.

In the past decade, investigation of the risk factors for lameness has been a major focus [33]. Such studies enable the identification of herd and cow level factors influencing the occurrence of clinical lameness and specific claw lesions [34]. Bruijnis et al. [7] suggested that improving farmers’ awareness of the causes and risk factors for lameness could motivate them to implement certain preventive measures. In one study, although the majority of the farmers surveyed were aware of such risk factors, they failed to adhere to the recommended measures to improve claw health [17]. Thus, farmers’ awareness of the potential risk factors may not necessarily translate into a reduction in on-farm lameness. Farmers often consider the costs and benefits of implementing any changes on their farms [23]. For farmers to implement on-farm changes, they need to be convinced of their capacity to effect the change and the efficacy of measures for lameness control [19]. By focusing on farm-specific risk factors, farmers could be more motivated to take decisive action. In the long term, such decisions are influenced by current lameness prevalence, as it will affect farmers’ perceived risk of future challenges [19].

### Implications of Farmers’ Perceptions and Underestimation of Lameness

Since farmers have the primary responsibility for ensuring the welfare for their animals, their perception of lameness is essential. This relates to the welfare implications of late lameness detection and treatment plans for specific claw lesions [5]. Accordingly, the majority of the farmers in a survey considered the same treatment plan for all lameness causes: trimming an affected claw with or without the application of a hoof block [29]. As farmers often reserved the term ‘lame’ for severely lame cows [5], it is suggested that cows with mild gait changes were not seen as lame. Such normalization of a problem, aside from increasing the risk of under-diagnosis, reduces its perceived gravity [4] (see Scheme 1). Hence, the mildly lame cows are less likely to be presented for treatment or maintenance hoof trimming.

The targeting of ‘early and effective treatment’ on mildly lame cows [35] has been demonstrated to enhance quick recovery, compared to cows that are chronically lame [36]. Additionally, moderately lame cows were recently reported to undergo marked behavioral changes in lying down, standing and feeding, which were presumed to be associated with severe lameness episodes [37]. The adverse impact of lameness on milk yield and reproductive performance has also been reported [10,38]. This is often linked to behavioral alterations, as found in the study mentioned [37], as well as to estrous activity resulting from painful claw lesions [39,40]. In a study where 75% of the cows with high LS were identified and treated by farm staff, only 35% of such treatments were carried out in less than three weeks after the first detection of lameness [41]. Aside from the welfare implications of such delayed treatment, economic losses also arise from lower milk yield and the additional management required (Scheme 1). For example, reports by Charfeddine and Cabral-Perez [10] showed the losses incurred from severely lame cows were three times greater than from the mildly lame group but the latter group still produced significant losses compared to sound cows. Further, studies to demonstrate the economic implications of delay from time of diagnosis to treatment are important to educate farmers on the need for proactive rather than reactive management strategies.

## 6. Farmers’ Participation in Studies Relating to Lameness Management

Lame cows are often not given immediate care and treatment, for various reasons [35,41]. Alawneh et al. [41] suggested that improving the ability of farmers to detect lame cows could reduce the incidence of delayed treatment. To that end, approaches to exemplify their potential in effecting the required management changes and to assess the efficacy of lameness management and treatment protocols are pertinent. Table 2 presents details of studies involving dairy farmers’ participation in the prevention and treatment of lameness in dairy cows.

Chapinal et al. [42] carried out a longitudinal study to assess the prevalence of lameness and hock lesions in 15 free-stall dairy herds in the United States. The enrolled farmers were educated periodically on how to improve claw health in their herds. The prevalence decreased significantly after the first and second assessments (herds with the highest prevalence at first visit). Again, results from another study involving two groups of farmers showed that those without support from a facilitator or veterinarian were less effective in implementing the recommended changes to reduce the risk of lameness [43]. The randomized control trial conducted by Groenevelt et al. [44] also reported a significantly higher response rate (reduction in LS) in cows that were treated by farmers compared to the control group receiving no treatment. In another study in which two groups of lame cows were identified and treated by trained personnel and farmers, lameness was significantly reduced in the former group, at the first week and final stages of the project [45].

The majority of the reviewed papers herein showed that farmers’ participation and education led to significant improvement in claw health and lameness reduction [43,44]. Nevertheless, while using a basic innovation diffusion model, Bell et al. [47] reported that over a one-year intervention period, the provision of validated external advice had a limited effect on lameness prevalence in primiparous heifers. The ‘health belief model’ described by Ritter et al. [19] may play a role in the findings above. The model suggests that the perceived risk associated with a health problem could be modified by knowledge interpretation. This relates to the farmers’ perception about the severity of lameness and the benefits of treating lame cows. These combined perceptions could affect the farmers’ decisions regarding lameness management strategies.

## 7. Factors Influencing Farmers’ Decision on the Management of Lame Cows

### 7.1. Perception Regarding Pain Management and Treatment

Pain management is a vital welfare aspect in lameness treatment [22,32]. An underlying pain may not be accompanied by apparent gait impairment, since behavioral changes to noxious stimuli could vary among animals [48]. However, adequate pain management remains crucial during both preventive and therapeutic CT [32,49]. The perception of the pain experienced by lame cows was the reason why some farmers felt the need for prompt treatment [22].

Previous studies relating to therapeutic CT showed that few farmers, veterinarians and claw trimmers used analgesics when treating lame cows [3]. Similarly, Becker et al. [22] investigated the attitudes of farmers to pain management during CT. They found that the respondents considered the administration of local anesthesia reasonable when they estimated that CT without analgesia would cause a greater level of pain [3]. Tensions between claw trimmers and veterinarians were suggested to influence farmers’ decisions on pain management. The availability of cheap labor and lower treatment cost offered by claw trimmers compared to veterinarians might encourage farmers to seek the formers’ services and give less consideration to pain management. Again, farmers who accorded less priority to the economic returns from dairy cows might pay more attention to an affected cow [50]. Cows producing little milk are at lower risk of developing claw horn disruptive lesion (CHDL), thus less liable to require pain management [22]. Hence, discussion between veterinarians and farmers should include pain management arising from treatment protocols, as well as the short- and long-term cost.

There are still limited detailed data on the treatment of CHDL. In a study by Bran et al. [21], most of the surveyed farmers stated they often treat lame cows using topical hoof products and antibiotics, whereas corrective CT and analgesics were rarely administered. This finding indicates the need for standardized dissemination of information to farmers, which could influence their decision-making on lameness management.

### 7.2. Farm Design and Management Systems

Farm design and management systems are essential housing factors associated with claw health and lameness in dairy herds and have the potential to influence farmers’ devising of lameness control plans. The farmers surveyed by Horseman et al. [5] indicated that farm layout, among other factors, hindered them from detecting lame cows. In the application of manual LS, multiple strides, undisturbed locomotion and the presence of a flat, non-slippery surface are fundamental considerations when scoring a cow for mobility [51,52]. However, stall designs may not support all the recommended provisions. Equally, recent reports suggested that automated lameness detection systems should be customized for farm-specific usage [11]. For example, in tie-stalls, unlike in free-stalls, cows seldom walk around. Hence, farms in the former may need to rely more on postural changes associated with lameness. Cutler et al. [17] observed that farmers in the tie-stall group made fewer management changes for lameness control in the previous two years than in the free-stall group.

Findings from studies on the occurrence of lameness and claw lesions either in pasture-based (grazing herds) or housed-cubicle dairy farms are also vital. With the reduced movement in housed indoor cows compared to those on pasture, there is a greater risk of lameness arising from painful claw lesions [53]. This phenomenon was observed among the pasture-based farmers surveyed by Olmos et al. [6] in southern Brazil, where claw disorders were not perceived as causes of lameness and preventive strategies such as CT were regarded as unnecessary. Similarly, trained personnel identified lame cows in grazing farms where the owners perceived that lameness had never occurred [20]. The overall effect on claw health may differ depending on other herd-level factors. For instance, housed cows provided with clean and dry surfaces, soft bedding and periodic CT, would enjoy healthier feet, despite being restricted from pasture access.

### 7.3. Herd Size and Time to Detect Lame Cows

The association between herd size and farmers’ actions on claw health has been reported in some studies [22,50]. In relation to lameness, visual LS, although easy to perform, could be time-consuming, especially in large dairies. Surveys have shown that lack of time to detect lame cows was a barrier to prompt treatment [4,5]. Examination of the hoof could be time-consuming, especially where facilities are lacking [35]. Automated lameness detection systems are alternatives to counter some of these challenges [11]. However, whether farmers prefer the cost-effective approach is unknown and requires further research [11]. Kielland et al. [50] proposed associations between herd size, expected economic gain and the attention given to dairy cows. Economic returns in smaller-sized herds might not be the priority, thus encouraging the farmer to pay more attention to the cows and accordingly to provide immediate care for lame cows. However, the situation can be the other way around. Irrespective of herd size, farmers may be willing to provide immediate care to highly productive cows as soon as they are detected lame.

### 7.4. Cultural and Socio-Economic Factors

Although the understanding and capacity of farmers to absorb information may vary; cultural, socio-economic and environmental factors are also influential. These areas are often ignored and displaced by inappropriate advice and actions or inaction [50]. Several studies have identified these factors as barriers to the uptake and success of various strategies to reduce lameness [4,6,46]. Measures to improve the prevalence of lameness in dairies need to consider the potential barriers inherent in heterogeneous communities having limited resources and a strong, enclosed ethnography [6].

### 7.5. Availability of Skilled Labour and Equipment

For effective management of lameness in dairy herds, the input from trained personnel such as veterinarians, dairy consultants and professional hoof trimmers is pertinent. In the absence of such expertise, various farm practices have been found to increase the risk of occurrence of claw lesions [43,54]. For instance, inappropriate CT was reported as a major cause of the increased prevalence of painful claw lesions such as toe ulcers [54]. The availability of such skilled labor varies from one region to another and may, regardless of the perceived risk of lameness, affect farmers’ actions in relation to lameness prevention or treatment. Adams et al. [55] found that, among 184 farms surveyed in the USA, only 7% were not practicing CT, while two-thirds and 16% used hoof-trimming services and on-farm staff or veterinarians, respectively. In contrast, the high prevalence of overgrown claws in dairy herds in Selangor, Malaysia, indicated that farmers in the region rarely practice CT [56].

Labor availability also affects the treatment of particular lameness causes, depending on the severity of the condition and the expected expenditure. The likelihood of needing labor to treat an individual cow depends on the specific causes of lameness, ranging from 20% for interdigital dermatitis (IDD) and sole hemorrhage (SH) to 100% for foot rot [55]. However, for treatment by hoof trimmer, the likelihood of treatment for specific lameness causes ranges from 0% for foot rot to 40% for SU. The lowest likelihood is that a veterinarian would be treating a case (ranging from 1% for WLD, SH, DD and HHE, to 5% for foot rot and SU) [55]. Difficulty in accessing the service of a claw trimmer or veterinarian for every case may contribute to this low probability of receiving treatment for a case from such a professional. Again, claw lesions that are difficult to treat may be left unchecked unless they are diagnosed when a skilled person is present. Claw trimmers may provide cheaper labor than veterinarians, which could prompt farmers to choose to hire the former [22].

Lame cows require additional care. As such, availability of equipment for the purpose has been shown to influence farmers’ decision regarding when and how lame cows are treated [4,35]. Farms with adequate restraining equipment, such as a hoof-trimming chute, are more likely to monitor cows for claw health and lameness, as well as devise treatment plans, compared with farms lacking such provisions.

## 8. Economic Aspects

The economic implication of lameness has gained more interest in the dairy industry [23]. Besides good welfare and animal health, profit maximization remains a primary target of dairy farmers [2]. Hence, studies to determine the economic losses accrued due to dairy cattle lameness and the benefits of expenditures on lameness preventive measures are important for farmers’ education and decision making.

Expenditure on disease can be looked at from the point of view of treatment or of prevention [2,23]. For instance, the required labor and supplies are parts of the treatment costs [2]. Accordingly, the farmer may pay for services rendered for treatment, whereas facilities and inputs such as lameness detection systems are regarded as preventive expenditures [2]. Certain practices such as CT could fall into either of the categories, as it could either be for preventive or therapeutic purpose.

### 8.1. Treatment Costs

Labour as an aspect of the treatment cost varies depending on the specific cause of lameness [2]. A study of 10 veterinarians in the United Kingdom showed that, although the total cost for veterinary labor per affected cow was lowest for the treatment of SU, it incurred the highest labor charge amounting to 66 USA dollar (USD) as at 1997 [57]. Accordingly, treatment of CHDL would require individual assessment and may vary among cows, unlike infectious claw lesions, which can be treated in a group. Additional labor may also relate to the specific treatment protocol, such as corrective trimming, block administration and pain management. These costs were also dependent on the severity (mild or severe) of the lameness conditions and the level of involvement of veterinarians and hoof trimmers in lameness management [10].

The costs attributed to therapeutics is another component of losses related to lameness. These costs were found by Dolecheck et al. [2] to range from 2 to 37% for the treatment of every case of lameness. Also, only a displaced abomasum had a higher expenditure (for the therapeutics used) than lameness among the dairy diseases investigated by Liang et al. [58]. CHDL, such as WLD and SU, were the highest contributors of therapeutics to the total cost per case (both mild and severe lesions) [10]. In Spanish dairy herds, the average annual cost for CHDL (SU and WLD) as at 2017 was estimated as 47 USD, whereas the annual extra expenditure for cows affected with SU and WLD was 3256 USD and 2765 USD, respectively [10]. These economic implications differ from one region to another and between herds, depending on the prevalence of claw lesions. These aspects need to be considered when educating farmers about strategies for lameness control.

### 8.2. Costs of Prevention

Preventive measures in relation to lameness are strategic plans to reduce or minimize the occurrence of claw and leg disorders. Demonstrating the returns from the expenditure and the time spent on productivity is pertinent in this respect. A study conducted in 2006 in Danish dairy herds showed that the highest and lowest contributions to the increased total profit margin/cow-year were the provision of rubber flooring (9 USD/cow-year) and foot bathing (2 USD/cow-year), respectively [59]. Similarly, in a study by Bruijnis et al. [60], improving the lying surface was the most cost-effective preventive measure, providing better welfare benefits in cases of lameness, followed by reducing stocking density and periodic CT. An approach termed “willingness to pay” (WTP) was used by Bennett et al. [61] to assess farmers’ attitudes towards lameness control and reduction. Van De Gutch et al. [11] note that “WTP is calculated as a measure for the amount of money that a farmer is willing to spend for certain improvements in the characteristics of a system.” A median WTP of £249 (Great Britain pound as at 2014) per lame cow was observed among the surveyed farmers [61]. However, they were not willing to experience the inconvenience associated with implementing lameness control measures and they seldom adopted those available to them.

Decision-making by farmers might include choosing between long- or short-term preventive measures. Measures such as CT, foot bathing, selection of cows based on claw traits and maintaining BCS could be seen as short-term investments in lameness prevention. On the other hand, changing stall designs and rubber flooring are long-term investments [2]. To motivate farmers concerned about economic return, the preventive costs associated with any management plan relating to lameness need to be elucidated.

### 8.3. Lameness Detection and Economic Value

A fundamental approach in lameness management is the routine evaluation of claw health and the detection of associated disorders [24]. Farmers need to be convinced of the best and most economically viable practice for the prompt and early detection of lame cows [2]. For this purpose, visual or manual LS systems are cost-effective, since they involve only labor expenditure, in the form of activity by the investigator and directing of the cows during locomotion. The constraints include the time taken to observe the cows in the herd and the subjectivity of the scoring technique [51,62].

Farmers have budgeted time for on-farm activities. As such, they often consider their economic advantage based on the trade-offs between one operation and another. The limitations of manual LS systems such as lower sensitivity to detect lame cows and the presence of lesions are major reasons why farmers attach lower economic value to their application [51]. In some herds, claw trimmers are given the sole responsibility to detect lameness. However, lame cows are missed between trimming periods, which may further increase the accrued losses.

Automated lameness scoring systems (ALSSs) have shown the potential to revolutionizes early lameness detection [51,62]. To farmers, such systems have economic value if the definite causes of lameness are easily detected. This will enable the provision of an appropriate treatment plan and justify input into the labor of restraining the cow and claw examination, as seen in the GAITWISE system [63]. Certain automated systems such as image-processing techniques require limited labor input [64]. While some ALSSs are cost-effective, the majority of the existing designs are expensive to install, and current farm designs may not support their function [65]. These are principal reasons for the little economic assessment of ALSSs that has been conducted in several studies, compared to other health conditions in dairy production [2]. From the farmers’ perspective, the complexities of lameness issues, the price of upcoming system models and performance uncertainties are factors to be considered [11].

## 9. Conclusions

The dairy farm needs to be viewed as integrated system in which the veterinarian contributes knowledge that translates into on-farm application. Educating and motivating farmers are important elements in ensuring the success of such a system. Lameness remains an important welfare and economic concern in the dairy industry. Based on the recent progress made in lameness-related research, the focus has moved from treatment to prevention. Nevertheless, behavioral changes among farmers and dairy consultants or veterinarians are necessary for the realization of such a paradigm shift and responsibility needs to be shared among them, underpinned by improved communication.

This review discusses the issues affecting dairy farmers’ actions in relation to lameness and their adoption of recommended preventive measures. The solutions could be tailored towards improving how farmers acknowledge the existence of the problem and their responsibility to act accordingly. Again, their perception of the feasibility and cost effectiveness of any recommended management strategy is vital. Farmers often weigh the expected returns from any recommended changes against the disadvantages. Hence, both the animal welfare and the economic implications of lameness are aspects that require more evaluation.

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
