# Peer review of "Dairy Farmers’ Perceptions of and Actions in Relation to Lameness Management"

_animals, 2019, doi:10.3390/ani9050270_

Round 1
Reviewer 1 Report
The present work lacks of clarity and flow. In general I recommend to review the manuscript particularly in the following points:
Aim: why did authors decided to carry out this review? what the reader will get from it? How this will benefit lameness research?
Materials and methods: this section needs to be explain clearly so it can be reproduced if needed. At the moment is not clear if more databases that the ones mentioned were used; what was the criteria inclusion, and why did authors decided to present the results under six main headings.
Fig 1: it is not clear what the diagram tries to show,and how this is link to the text. Was this diagram built with information from a paper? then reference is missing.
Tables: "selected studies" is this referring to the studies that passed the inclusion criteria? Titles need more clarity on what the tables are about. Table 1 is missing heading reference.
Sections talk about "several papers" how many of the papers reviewed were used in each topic. This information is important to have a picture of how much research has been done in each area.
Citing style: references are located in the middle of sentences and in some points there is no reference to support arguments or facts.
Overall it is recommended that authors provide a critical assessment of the literature presented, why did they selected the 40 papers, how these papers fall within the categories, why these categories, does this review provide information on a gap in the literature, what is the value for the reader.
Author Response
Dear Reviewer,
Response is as attached for your perusal.
Regards,
Siti

Reviewer 2 Report
Dear Authors,
General comments:
How dairy farmers percive lameness and how they are motivated to take action is a highly relevant topic.
I find the paper to be well structured and interesting, however I believe it would benefit from a close editing and proof reading by a native speaker of English.
Specific comments:
Line 55: I recommend that reference 4 is moved to line 57
Line 58: Make sure you use the same tense in the sentense (view=present, had=past)
Line 59: Please rewrite the sentense starting with 'For instance, farmers...', so that it is clear to the reader that you are referring to farmers included in a particular study (e.g. 'For instance, in a study by Olmos, farmers were...').
Line 61: Same as above.
Lines 86-88: I find it difficult to understand this. What is meant by the primary sources and the primary database?
Line 89: Can you provide a reference for the PRISMA checklist?
Line 111: I suggest 'less' instead of 'little'.
Line 115: Same as line 111.
Lines122-123: Same as line 59.
Line 123: ...to take action...
Line 151, Table 1: Why are the two columns to the right not split all the way down? For instance, Espejo mentions the mean estimate by farms manager (8.3%), this should be included for clarity.
Line 157: Same as line 59
Line 175: Should it read 'terms' instead of 'items'? If not, I don't understand the sentense.
Line 180: It is unclear how this reference (31) relates to this sentense.
Line 203: Consider rewriting: 'Since farmers have the primary responsibility...'
Line 237, Table 2: You use both 'TG' and 'TX' in the table, but in the keys only 'TX' is mentioned.
Line 270: I believe the correct reference here is no. 20, Becker.
Lines 316-317: Is this you opinion or do you have a reference to back this statement?
Lines 332-334: This sentense is difficult to read and understand. Please rewrite, perhaps split it in two.
Lines 349-351: I disagree with the sentence: 'However, in countries...'. Is it your opinion or do you have a reference to back the statement?
Lines 394-395: Same as line 59 (e.g. 'In a study by Bennett, ...').
Line 415: Consider rewriting: 'are given the sole responsibility...'
Author Response
Dear Reviewer,
Response is as in the attached document.
Thanks and regards,
Siti

Round 2
Reviewer 1 Report
Dear authors,
Thank you for considering the comments. However, the manuscript still needs a language revision and style editing. Please see below for comments and suggestions. These are just some examples of style and grammatical corrections that need to be carried out to improve the quality of this manuscript.
Abstract
L28. Sentence starting “Minimizing these problems…” please delete or improve
L 31. Change “which results …” to “as a result of different understanding of the problem”.
L31-32. Modify the sentence “The same applies to…” to “The same applies to their decision to treat lame cows and to adopt various detection ….”
L34. Modify “Overall, these events” to “All of these…”
L34. Modify “suboptimal” to “poor cattle”
L38. Change “to buttress” to “to strength”
L39. Change “countering” to “answering”
L40. Change “prospects” to “opportunities”
Introduction
L46. Lameness causes indeed pain and behaviour changes in dairy cattle, it does not limit production it affects milk production due to the stress caused to the animals. Also, it is not an economic problem in dairy cows, it is an economic problem for milk production systems. I suggest rewriting this sentence, so the message is clear.
L47. The occurrence of lameness?
L48. Delete “investigating”. This sentence is missing references, what are these papers?
L49. Have highlighted the importance?
L51 Please improve this sentence
L54 Add “Then” before “effective communication”. In this sentence, “these personnel” please change to “them”
L56 please add “s” to "constitute"
L59. Please provide all the references that you used to state this point: Such differing perception is influenced by several factors.
L60. Please add “an” before “optimal”
L66. “incident” - do you refer to incidence of chronically lame cows?
L67. Please improve this sentence.
L68. Add “s” to “include”
L70 add “the” before treatment
L75. Add “that” after “current data” and delete “to”. Also summarize is missing an “r”
L76 please change to “influencing their lameness management practices”
L77 change “personnel” for staff
L78 change “motivating” for “to motivate”, and “in implementing” for “to implement”
L79 please delete “present” and add “the” before “literature”
L80 Is the objective of the method or is the objective of the manuscript? Also, could you clarify if the “suggested ways” is coming from the reviewed material or from your conclusions from the reviewed material. Please modify this sentence so your message is clear.
L85 Add “a” before “systematic”
L85 Bias instead of biaz
L86 Fulfil instead of fulfil
L88 delete “year”
L89 Please join these two paragraphs.
L90 delete “,” before 2018
L91 Scholar is with capital letter
L98 delete “namely” redundant. Add “:” after “headings”. Add “The” before “importance”
L99 You talk that there are six main headings, but the ones given here do not match with the actual headings below. Probably these are six topics? Still, make sure that the headings or topics here match what you are describing below.
L108 I am not sure if I understand this sentence, participants had higher preference … difficulty in calving? Please review this sentence as it is not clear.
L121 – 127 This paragraph sounds like other diseases prevalence affect farmers’ perception on lameness and claw health. Is this correct? If so, then the introduction sentence should be changed to make this point clear.
L128 This sentence is not clear. I guess you mean that the welfare of lame cows is what influences farmers’ perception.
L129 Same with the sentence in this line. The welfare implications are for the cow, who feels pain. Please correct.
L134 Please correct this sentence, it is not clear.
L136-139. This paragraph seems loose here. It is not strong and the section about economics is the last one. I would recommend integrating this paragraph to that section.
L141 This sentence needs editing
L142 Change “is” for “are” before “fundamental”
L143 Not sure if you mean that farmers have reported that they underestimate lameness prevalence or studies have reported that farmers underestimate lameness prevalence. Please correct this sentence accordingly.
L144 Please correct this sentence as it is not easy to read.
Please make sure you continue with the language and style correction from these point forward.
Author Response
Dear Reviewer,
attached is our reply to your recent reviwer report.
We thank you very much.
